# Molecular Profiling Reveals Characteristic and Decisive Signatures in Patients after Allogeneic Stem Cell Transplantation Suffering from Invasive Pulmonary Aspergillosis

**DOI:** 10.3390/jof8020171

**Published:** 2022-02-10

**Authors:** Tamara Zoran, Bastian Seelbinder, Philip Lewis White, Jessica Sarah Price, Sabrina Kraus, Oliver Kurzai, Joerg Linde, Antje Häder, Claudia Loeffler, Goetz Ulrich Grigoleit, Hermann Einsele, Gianni Panagiotou, Juergen Loeffler, Sascha Schäuble

**Affiliations:** 1Department of Internal Medicine II, University Hospital Wuerzburg, 97080 Wuerzburg, Germany; Zoran_t@ukw.de (T.Z.); Kraus_S3@ukw.de (S.K.); Loeffler_C@ukw.de (C.L.); Grigoleit_G@ukw.de (G.U.G.); Einsele_h@ukw.de (H.E.); 2Systems Biology and Bioinformatics Unit, Leibniz Institute for Natural Product Research and Infection Biology—Hans Knoell Institute, 07745 Jena, Germany; bastian.seelbinder@leibniz-hki.de (B.S.); gianni.panagiotou@leibniz-hki.de (G.P.); 3Public Health Wales, Microbiology Cardiff, UHW, Cardiff CF14 4XW, UK; lewis.white@wales.nhs.uk (P.L.W.); Jessica.Price2@wales.nhs.uk (J.S.P.); 4Research Group Fungal Septomics, Leibniz Institute for Natural Product Research and Infection Biology—Hans Knoell Institute, 07745 Jena, Germany; okurzai@hygiene.uni-wuerzburg.de (O.K.); antje.haeder@leibniz-hki.de (A.H.); 5Institute for Hygiene and Microbiology, Julius Maximilians University of Wuerzburg, Josef-Schneider-Straße 2/E1, 97080 Wuerzburg, Germany; 6Friedrich—Loeffler Institute, Institute of Bacterial Infections and Zoonoses, 07743 Jena, Germany; joerg.linde@fli.de

**Keywords:** host response, invasive pulmonary aspergillosis, alloSCT patients, galectin-2, caspase-3, matrix metallopeptidase-1

## Abstract

Despite available diagnostic tests and recent advances, diagnosis of pulmonary invasive aspergillosis (IPA) remains challenging. We performed a longitudinal case-control pilot study to identify host-specific, novel, and immune-relevant molecular candidates indicating IPA in patients post allogeneic stem cell transplantation (alloSCT). Supported by differential gene expression analysis of six relevant in vitro studies, we conducted RNA sequencing of three alloSCT patients categorized as probable IPA cases and their matched controls without *Aspergillus* infection (66 samples in total). We additionally performed immunoassay analysis for all patient samples to gain a multi-omics perspective. Profiling analysis suggested *LGALS2*, *MMP1*, IL-8, and caspase-3 as potential host molecular candidates indicating IPA in investigated alloSCT patients. MMP1, IL-8, and caspase-3 were evaluated further in alloSCT patients for their potential to differentiate possible IPA cases and patients suffering from COVID-19-associated pulmonary aspergillosis (CAPA) and appropriate control patients. Possible IPA cases showed differences in IL-8 and caspase-3 serum levels compared with matched controls. Furthermore, we observed significant differences in IL-8 and caspase-3 levels among CAPA patients compared with control patients. With our conceptual work, we demonstrate the potential value of considering the human immune response during *Aspergillus* infection to identify immune-relevant molecular candidates indicating IPA in alloSCT patients. These human host candidates together with already established fungal biomarkers might improve the accuracy of IPA diagnostic tools.

## 1. Introduction

Invasive pulmonary aspergillosis (IPA) is often associated with a high mortality rate as a result of late diagnosis due to frequent, nonspecific symptoms and the limitations of currently used diagnostic tools including fungal biomarkers. Diagnosis of IPA is based on the integration of radiological, clinical, and microbiological data [1]. In the current guidelines of the European Organization for Research and Treatment of Cancer and the Mycoses Study Group (EORTC/MSG), microbiological evaluation includes conventional methods such as culture and microscopy examination from bronchoalveolar lavage (BAL) fluid or biopsies along with galactomannan antigen and *Aspergillus* DNA detection in BAL or blood [2]. Currently, detection of the cell wall component galactomannan with a commercially available enzyme-linked immunosorbent assay (ELISA) and detection of *Aspergillus*-specific DNA by polymerase chain reaction (PCR) are essential tools for IPA diagnosis [1,2,3]. Despite these advances and different available assays, diagnostic tools for identifying IPA provide variable sensitivity and specificity. The ambiguous classification of possible IPA or the presence of positive mycology without supporting radiology typical of IPA represents a major challenge for clinicians when deciding on appropriate management [2]. Given the frequency that these cases are encountered in the clinic, there is a clear need for an unambiguous diagnosis of aspergillosis.

To date, human biomarkers for diagnosing IPA, including monitoring treatment response and patient outcomes, have been under-represented in studies evaluating IPA diagnostics [4]. Discovery and validation of novel human biomarkers are challenging due to the complexity of patients suffering from hematological malignancies, given their underlying disease, frequent stem cell transplantation and associated GvHD, co-infections with various microorganisms (such as CMV and severe influenza) [5,6], and subsequent treatment, including corticosteroid and immunosuppressive therapy. Furthermore, aspergillosis has been observed as a complication in severe coronavirus disease 2019 (COVID-19) patients and was responsible for increased mortality [7]. Several European medical centers have reported COVID-19-associated pulmonary aspergillosis (CAPA) in 20–35% of mechanically ventilated patients [8].

Acknowledging the benefit of personalized medicine, this longitudinal case-control pilot study aimed to identify immune-relevant gene or protein candidates indicative of ongoing IPA in alloSCT patients, which would increase the diagnostic specificity. In addition, we aimed to explore if an integrative analysis of different parameters might be feasible for improved stratification and characterization of alloSCT patients suffering from IPA. We report a characteristic signature (*LGALS2*-*MMP1*-Caspase-3) identified by an integrated approach based on differential gene expression analysis of seven relevant gene expression datasets, combined with transcriptome and protein profiling of three selected IPA cases and their three matched controls with no evidence of *Aspergillus* infection. Selected molecular candidates were further evaluated by qPCR and immunoassays in additional alloSCT patients categorized as probable and possible IPA cases [2] as well as COVID-19 patients who developed aspergillosis as a secondary infection (CAPA patients), matched to the appropriate control patients.

## 2. Materials and Methods

### 2.1. Study Design and Patient Recruitment

In this longitudinal, case-control pilot study, patients at the University Hospital of Würzburg (Würzburg, Germany) who received allogeneic stem cell transplantation (alloSCT) for hematological malignancies from March 2017 to December 2019 were recruited and categorized according to the time-actual EORTC/MSG criteria [2]. To identify characteristic signatures indicating IPA, we prospectively collected human blood samples from 90 patients undergoing alloSCT in a biweekly manner until 100 days after alloSCT. Three of these patients had microbiological (positive galactomannan test and/or *Aspergillus* PCR) and radiological evidence of *Aspergillus* infection along with host factors (neutropenia) and were categorized as probable IPA cases according to EORTC/MSG criteria [2]. Case and control patients were matched by the same sex and underlying disease (acute myeloid leukemia, AML), a similar age (at most a 9-year difference), and the time since receiving alloSCT (Table 1). Since patients developed probable IPA at different times after they received the alloSCT, we established a relative time point zero for the onset of probable IPA. Day 0 represented the first positive *Aspergillus* PCR or galactomannan test either in serum or BAL together with radiological evidence (CT scan) suspecting fungal infection. Furthermore, all six patients developed neutropenia after alloSCT. Clinical characteristics of alloSCT patients selected for profiling are shown in Table 1. From these selected patients, we collected blood samples, which were retrospectively investigated on the transcriptome and protein level (see Section 2.2, Section 2.3 and Section 2.4). In addition, we investigated protein levels of molecular candidates by immunoassays (see Section 2.4) in serum samples (four consecutive time points per patient) of possible IPA cases (*n* = 3) and controls (*n* = 3) (Appendix A).

Additional patient cohorts were obtained from Public Health Wales, Microbiology Cardiff (Wales, United Kingdom), termed Cardiff I and Cardiff II, and used to further evaluate selected molecular candidates using immunoassays (see Section 2.4. Protein quantification in serum). The Cardiff I cohort consisted of additional alloSCT patients suffering from probable IPA (*n* = 6), possible IPA (*n* = 5), and control patients (*n* = 6). From these patients, 5–6 serum samples from consecutive time points after IPA onset were investigated. The Cardiff II cohort comprised 65 COVID-19 patients; among those, 20 COVID-19 patients suffered from CAPA [9]. Clinical characteristics of additional patients included in our study are shown in Appendix A.

This study was approved by the Ethics Committee of the University of Würzburg, Würzburg (ethics approval number: #173/11 and #225/12). All sera from Cardiff were surplus clinical samples retrospectively and anonymously tested by selected protein immunoassays. Prior to testing, NHS Health Research authority decision tools were utilized, deeming that ethical review was not required.

### 2.2. Blood Sample Collection and Processing

After allogeneic stem cell transplantation, blood samples were collected twice weekly, whenever feasible, until day 100 after transplantation. In the case of readmission to the hospital, the collection of blood samples started again. Blood was collected according to the instructions of tube providers. Serum was collected into SARSTEDT Monovette^®^/9 mL tubes (Sarstedt, Nümbrecht, Germany) and centrifuged at 3000× *g* for 10 min. Whole blood was collected in Tempus™ Blood RNA Tube RNA preservation tubes (Thermo Fisher Scientific, Waltham, MA, USA) and immediately mixed vigorously for at least 10 s. Serum samples were aliquoted and frozen together with Tempus tubes at −20 °C until further analysis.

### 2.3. Whole-Blood Transcriptome Profiling

Total RNA from whole blood, collected in Tempus tubes, was isolated with a Tempus™ Spin RNA Isolation Kit (Thermo Fisher Scientific, Waltham, MA, USA) and treated with DNase using a TURBO DNA-free™ Kit (Thermo Fisher Scientific, Waltham, MA, USA) following the manufacturer’s protocol. RNA purity and concentration were investigated via a NanoDrop ND-1000 spectral photometer (Thermo Fisher Scientific, Waltham, MA, USA). The integrity of RNA was determined with a 2100 Bioanalyzer (Agilent Technologies, Waldbronn, Germany) using RNA 6000 Pico or Nano LabChip Kits (Agilent Technologies, Waldbronn, Germany) according to the manufacturer’s instructions. RIN values of investigated samples were in the range of 7.6–9.9 and no ribosomal 18S or 28S peaks were detected in the profiles.

RNA sequencing libraries were generated with the Illumina TruSeq^®^ Stranded mRNA, including the Ribo-Zero Globin technology that depletes cytoplasmic and mitochondrial rRNA and globin mRNA. RNA sequencing was performed by IMGM Laboratories GmbH (Martinsried, Germany) on the Illumina NextSeq^®^ 500 next-generation sequencing system with 1 × 75 bp single-read chemistry. Raw files are accessible under the Gene Expression Omnibus accession number GSE174825.

#### 2.3.1. RNA-seq Data Processing

Preprocessing of raw reads, including quality control and gene abundance estimation, was performed with the GEO2RNAseq pipeline (v0.100.1) [10] in *R* version 3.5.1. Quality analysis was performed before and after trimming with FastQC (v0.11.7). Read-quality trimming was performed with Trimmomatic (v0.36) [11]. Adapter sequences were removed, window size trimming was performed (15 nucleotides, average Q < 25), 5′ and 3′ clipping for any base with Q < 3, and sequences shorter than 30 nucleotides were removed. Reads were mapped against the human reference genome (GRCH 38, v109). First, the reference genome was indexed with exon information using HiSat2 (v2.1.0). Then, paired-end read alignment was performed using HiSat2 on the exon-indexed reference genome. SAMtools was used for indexing of mapping files and to obtain further statistical information. Only concordantly aligned pairs of reads were used. Mapping statistics were calculated using the “calc_mapping_stats” function of GEO2RNAseq. Gene abundance estimation was performed with featureCounts (R package Rsubread, v1.34.0) in paired-end mode with default parameters. MultiQC (v1.7) was used to summarize the output of FastQC, Trimmomatic, HiSat, featureCounts, and SAMtools (Appendix A). Count matrices were normalized using median-by-ratio normalization (MRN) as described before [12]. Principal component analysis was performed based on MRN gene-abundance data. Hierarchical clustering was performed with the ward.D2 clustering method using MRN gene-abundance data with pheatmap (v1.0.12).

Differential gene expression was analyzed by GEO2RNAseq. We used all available samples from patients developing probable invasive aspergillosis after allogeneic stem cell transplantation and compared them with selected samples of allogeneic stem cell transplantation control patients who did not develop invasive aspergillosis. The statistical tools DESeq2 (v1.18.1) and edgeR (v3.20.7) were used to report significantly differentially abundant genes. For each tool, *p*-values were corrected for multiple testing using the false discovery rate method (q = FDR(p)). In addition, mean MRN, transcripts per kilobase million (TPM), and reads per kilobase million (RPKM) values were computed per test per group, including the corresponding log_2_ fold-changes. Gene expression differences were considered significant if they were reported significant by both tools (q < 0.05 and |log_2_ MRN| ≥ 0.8) unless stated otherwise.

#### 2.3.2. qPCR-Based Validation

To validate the sequencing results of the profiled patient cohort (Table 1), selected molecular candidates were investigated by qPCR. RNA was transcribed in Thermo Cycler (Eppendorf, Hamburg, Germany) using a First Strand cDNA Synthesis Kit (Thermo Scientific™, Waltham, MA, USA) according to the manufacturer’s instructions. Based on the RNA-seq and previous in vitro studies, gene expression of the following molecular candidates was investigated using TaqMan Assays^®^ (Thermo Fisher Scientific, Waltham, MA, USA): LGALS2 (Hs00197810_m1), MMP1 (Hs00899658_m1), MMP9 (Hs00957562_m1), ITGB3 (Hs01001469_m1), SRSF4 (Hs00900675_m1), and TAOK3 (Hs00937694_m1). SRSF4 and TAOK3 were used as reference genes and were selected according to our sequencing results. TaqMan™ Fast Advanced Master Mix (Thermo Fisher Scientific, Waltham, MA, USA) was used for amplification of selected targets according to the manufacturer’s instructions. PCR was performed using an optical 96-well reaction plate (Applied Biosystems, Brighton, UK) in the StepOnePlus™ Real-Time PCR System (Applied Biosystems, Brighton, UK). Gene expression was analyzed and calculated according to the ΔΔCq method [13,14]. Time points of control patients were matched to the closest investigated time point of probable IPA cases.

#### 2.3.3. In Silico Investigation of the Gene Expression of Characteristic Immune-Relevant Candidates in Other Studies

To investigate host response during *Aspergillus fumigatus* infection, we additionally analyzed gene expression datasets of six in vitro studies using different immune cells ([15], GEO datasets GSE174825, GSE60729, GSE69723, GSE134344, and GSE177040). The investigated datasets were selected based on the infection model (immune cells infected with *A. fumigatus*) as well as at least 1000 identified differentially expressed genes obtained by either RNA sequencing or microarrays. Differentially expressed genes were either extracted from the publication’s supplementary files [15], computed with the web application GEO2R [16] with standard parameter settings for microarray data (GSE60729, GSE69723), or using again the GEO2RNAseq pipeline [10] for RNA sequencing data (GSE174825, GSE134344, GSE177040).

### 2.4. Protein Quantification in Serum

Immunoassays were performed to quantify the protein levels of selected characteristic molecular candidates in patient sera. The selection of analytes was based on transcriptome profiling results and previous studies [17,18,19]. Custom-made multiplex immunoassays (ProcartaPlex, Thermo Fisher Scientific, Waltham, MA, USA) were used to investigate the following analytes: Caspase-3, CD40, IL-8, MMP1, PAI-1, and VEGFA. Serum levels of ITGB3 (Abbexa, Cambridge, UK), LGALS2 (Abbexa, Cambridge, UK), and MMP9 (BioLegend, San Diego, CA, USA) were measured using singleplex immunoassays according to the manufacturer’s protocol. All data were analyzed following the manufacturer’s instructions. Additionally, an independent patient cohort consisting of COVID-19 patients with or without aspergillosis was investigated by the following immunoassays according to the manufacturer’s instructions: IL-8 (BioLegend, San Diego, CA, US), Caspase-3 (ThermoFisher, Waltham, MA, USA), and MMP1 (ThermoFisher, Waltham, MA, USA).

### 2.5. Statistical Analysis

To avoid patient sampling and time point bias, samples of the original investigated patient cohort (Table 1) were balanced for time points and patients. Unless stated differently, the first five weeks after IPA onset were used for analysis (Appendix A). Data were further analyzed using *R* version 3.5.1 and/or GraphPad PRISM (v6.0). If not stated otherwise, significant differences between investigated groups were tested using the Mann–Whitney test. Statistical significance between investigated groups is indicated as follows: *, *p* < 0.05; **, *p* < 0.01; ***, *p* < 0.001; ****, *p* < 0.0001.

## 3. Results

The workflow of the present longitudinal study, investigating the host response of alloSCT patients suffering from IPA, is shown in Figure 1. We used an integrated approach combining profiling of selected alloSCT patients along with investigation of seven gene expression datasets obtained from six relevant in vitro studies (cf. Section 2) studying host response following *A. fumigatus* infection. Protein levels of potential molecular candidates indicating IPA in the investigated alloSCT patients were assessed in an independent cohort consisting of COVID-19 patients suffering from aspergillosis as a secondary infection.

### 3.1. LGALS2 and MMP1 Are Indicative for IPA in alloSCT Patients within the First Five Weeks after Disease Onset

Our whole-blood sequencing results reveal characteristic immune-relevant gene clusters with a stable expression after IPA diagnosis over the investigated time (Figure 2a). Despite observed patient variability in gene expression within the investigated patient groups, e.g., with respect to gender, transcriptome profiling revealed characteristic gene expression profiles among probable IPA and control patients. Overall, our data indicate 1131 differentially expressed genes (DEGs) in IPA patients compared with their matched controls taking all investigated samples into account (Wilcoxon test; q < 0.05; log_2_FC cutoff 0.8) (Appendix A). Gene ontology enrichment analysis of the identified gene clusters revealed categories related to metabolic and cell motility processes as well as immune response and defence against fungi (neutrophil-mediated immunity, regulation of NF-κB transcription activity, activation of NK-cells, andT-cells and antibodies production). These enriched molecular functions showed that several host immune response-relevant gene expression patterns discriminate between IPA cases and matched controls (Figure 2a).

To identify characteristic molecular candidates indicating IPA after alloSCT, we investigated the RNA-sequencing-based differential gene expression of three probable IPA cases and their matched controls (*n* = 66) together with the differential gene expression of six additional and carefully selected in vitro studies and explored the existing literature for further potentially relevant genes in the context of mold infections [15,18,19,20,21,22,23]. Based on this combined analysis, we determined nine characteristic immune-relevant candidates for further evaluation: MMP1, MMP9, LGALS2, ITGB3, VEGFA, CASP3, CD40, CXCL8, and PAI-1 (Figure 2).

Since early and correct treatment is particularly critical for patient survival [24], we considered the first five-week period after the onset of IPA for further analyses unless noted otherwise. Furthermore, to avoid bias from individual patients, we balanced the number of patient samples by including four samples per patient across the first five weeks after individual IPA onset. Investigating the mean expression differences for selected molecular candidates, across the first five weeks after IPA onset, we observed significant induction in the expression of *ITGB3*, *MMP1,* and *MMP9* in probable IPA cases, while the expression of *CD40*, *CXCL8,* and *LGALS2* was significantly decreased compared with control patients (Figure 2b). Both the investigation of gene expression across all samples and the use of our balanced dataset revealed galectins, matrix metallopeptidases, and integrins to be robust potential characteristic molecular candidates indicating IPA, exemplified by *LGALS2*, *MMP1*, *MMP9*, and *ITGB3* (Figure 2a, right column).

For confirmation, we evaluated the promising molecular candidates (*LGALS2*, *MMP1*, *MMP9*, and *ITGB3*) independently by qPCR (Figure 3a). We confirmed significantly higher expression of *MMP1* (*p* < 0.0001), *MMP9* (*p* = 0.0173), and *ITGB3* (*p* < 0.0009) as well as significantly lower *LGALS2* expression (*p* < 0.0001) in probable IPA patients compared with control patients (Figure 3a). To explore gene expression over a longer time frame, we additionally investigated the gene expression levels of the selected candidate genes in profiled alloSCT patients using all profiled patient samples (up to 56 days after IPA onset) (Figure 3b).

Within the first two weeks after IPA onset, expression of *MMP1* was induced in almost all IPA cases compared with their controls. One matched patient pair (P15—IPA with P53—control) showed a decrease in the log_2_ fold change (Log2FC) of *MMP1* after two weeks and remained downregulated at subsequent time points, indicating some degree of expression variance after the first two weeks of IPA (Figure 3b). Moreover, we observed consistent downregulation of *LGALS2* expression up to 56 days after IPA onset in all probable IPA patients compared with their matched controls (Figure 3b). Although the log2FC of case-control-matched *MMP9* and *ITGB3* expression showed patient-specific variance, our results suggest *ITGB3* induction among the investigated IPA cases in the early phase after IPA onset. We observed a similar trend in *MMP9* expression in two investigated patients (P15 and P43) (Appendix A). These data suggest that *MMP1* and *LGALS2* have robust gene expression levels among the investigated patients, particularly during early IPA.

In addition to transcriptome profiling, we investigated publicly available relevant gene expression datasets studying the host response of different immune cells infected with *A. fumigatus*. We screened six in vitro studies (seven datasets) with at least 1000 identified DEGs based on both microarray and RNA sequencing and investigated the expression of the nine previously selected candidates (see Section 2). Investigation of these additional expression datasets indicated that all candidates were identified in at least one of the investigated datasets: *CASP3, CD40, CXCL8, ITGB3, LGALS2, MMP1, MMP9, SERPINE1,* and *VEGFA* (Appendix A). Interestingly, the following genes were found across all six investigated studies: *CD40*, *CXCL8*, and *MMP1*. Furthermore, *MMP1* induction was observed in all seven investigated datasets, while *LGALS2* downregulation was observed in two studies using monocyte-derived dendritic cells. These results provide further evidence supporting our transcriptome-based findings, particularly for *MMP1* and *LGALS2*, which warrant further evaluation in a larger patient cohort.

### 3.2. MMP1, IL-8, and Caspase-3 Serum Protein Levels Show Distinctive Patterns in Probable IPA Cases

Next, we investigated whether the identified characteristic molecular candidates (Figure 2b) also show distinctive protein abundance patterns in sera of investigated alloSCT patients. Data analysis based on all samples (*n* = 66) demonstrated significantly lower ITGB3 and higher serpine-1 serum levels in cases compared with controls, yet these differences were not clinically important (Appendix A). Interestingly, in contrast to a significant difference in the gene expression level for *LGALS2*, its gene product galectin-2 showed no significant difference in the protein level of probable IPA cases compared with controls, irrespective of the analyzed time frame (Figure 2b, Appendix A). Furthermore, CD40 was not detected across the majority of the investigated patient sera from our original Würzburg cohort; hence, these four candidates were not followed further.

To evaluate the robustness of our findings, we further quantified MMP1, IL-8, caspase-3, MMP9, and VEGFA in patient sera obtained from two independent medical centers: the University Hospital of Würzburg (original cohort) and Public Health Wales Microbiology Cardiff medical center (Cardiff I cohort) (Appendix A). The Cardiff I cohort comprised six probable IPA patients and their six matched controls. We analyzed six time points after alloSCT for each patient, in total 72 serum samples (Appendix A). While the Cardiff I cohort demonstrated significantly lower VEGFA (*p* = 0.0473) serum levels in IPA cases compared with controls, this trend was not observed in the original Würzburg cohort. In contrast, this cohort indicated significantly higher VEGFA levels in cases compared with controls (*p* = 0.0172), however only when all samples were analyzed (*n* = 66) (Appendix A). MMP9 serum levels were significantly elevated in IPA cases compared with controls irrespective of the analyzed time frame in the original cohort only, however without clinical importance (Appendix A).

MMP1, IL-8, and caspase-3 demonstrated the most promising results, indicating IPA in investigated alloSCT patients (Figure 4). IL-8 serum levels were significantly elevated in cases compared with controls in both the original Würzburg (*p* = 0.0165) and Cardiff I (*p* < 0.0001) patient cohorts. Additionally, caspase-3 indicated significantly higher levels in cases compared with controls in both patient cohorts (Würzburg, *p* = 0.0029; Cardiff I, *p* < 0.0001; Figure 4). Moreover, we observed significantly elevated MMP1 levels in cases compared with controls in the original Würzburg cohort (*p* = 0.0001). The Cardiff I cohort demonstrated a similar trend, yet this difference was not significant (*p* = 0.088) (Figure 4a,b).

Investigation of MMP1 protein levels on the individual level in our originally profiled Würzburg cohort up to 56 days after IPA onset revealed elevated levels in IPA cases compared with controls at almost all time points (Figure 4c). However, protein levels of IL-8 and caspase-3 did not show a clear day-by-day separation between cases and controls across all investigated 56 days on the individual level, highlighting patient and sex-specific patterns (Figure 4c). While our data demonstrate higher IL-8 and caspase-3 serum levels in the female IPA case (P43) and to some extent also in its control (P18), this trend was not observed in all female patients of the Cardiff I cohort (Appendix A).

Taken together, despite observed differences due to heterogeneic patient origin IL-8, caspase-3, and, particularly, MMP1 showed pronounced protein profile differences between the investigated cases and controls across most, albeit not all, patient samples.

### 3.3. Possible IPA Cases Revealed Higher IL-8 and Caspase-3 Protein Levels Compared with Control Patients

Possible IPA has a clear need for improved diagnostic confidence and studies characterizing the host response of possible IPA patients are under-represented. Therefore, we further investigated the identified potential molecular candidates indicative of IPA on the protein level by immunoassays in possible IPA cases and their controls [2] in the Würzburg and Cardiff I cohorts. Investigation of MMP1, IL-8, and caspase-3 serum levels in cases and controls showed no significant differences for MMP1. Significantly higher serum levels were observed for IL-8 (*p* < 0.0001) and caspase-3 (*p* < 0.0001) in possible IPA cases compared with controls in the Cardiff I cohort only (Figure 5). Thus, IL-8 and caspase-3 in particular showed center-specific serum levels, which highlights the ambiguity of the possible category and warrants further investigation.

### 3.4. Patients with CAPA Have Significantly Lower IL-8 and Caspase-3 Serum Levels Compared with COVID-19 Patients

Finally, we assessed promising molecular candidates indicating IPA in alloSCT patients using immunoassays in a non-classical COVID-19 cohort (Cardiff II). MMP1, IL-8, and caspase-3 protein levels in 20 patients suffering from CAPA and 45 control COVID-19 patients without *Aspergillus* infections were analyzed (in total 65 patients, 65 samples; Appendix A). While no significant difference was observed between CAPA and COVID-19 control patients in MMP1 serum levels, our data reveal significantly lower IL-8 (*p* < 0.0001) and caspase-3 (*p* < 0.0001) serum levels in CAPA patients compared with COVID-19 control patients (Figure 6a). Furthermore, we observed higher IL-8 and caspase-3 serum levels in male compared to female control COVID-19 patients (Figure 6b,c). In detail, male CAPA patients had significantly lower IL-8 (*p* = 0.0001) and caspase-3 (*p* = 0.0004) serum levels compared with COVID-19 control patients, whereas significantly lower levels in cases compared with controls in female patients were observed only for IL-8 (*p* = 0.0409), potentially due to limited sample numbers.

## 4. Discussion

The present case-control study focused on alloSCT patients as a complex model of high-risk and severely ill patients. We aimed to evaluate the feasibility of biomarker screening in alloSCT patients and identify candidates showing potential for further evaluation in a larger patient cohort. Additionally, we investigated whether our identified molecular candidates give additional benefit over fungal biomarkers alone during the onset of IPA and proposed an integrative biomarker-based strategy to improve IPA diagnosis warranting evaluation in a larger conceptualized study.

Despite advances in the field of fungal diagnosis and treatment in the past decade, diagnosis of IPA continues to be challenging. Poor outcome of IPA is associated not only with late diagnosis due to the lack of optimal diagnostic methods and usually unspecific symptoms of the disease [25,26], but also due to emerging antifungal resistance [27]. In recent years, the success of emerging new therapies for malignant and autoimmune diseases led to new risk groups for IPA without traditional risk factors. Cases of aspergillosis have been increasingly reported in patients receiving monoclonal antibodies [28,29], small-molecule protein kinases inhibitors [30,31], agents for autoimmune disorders, and novel immunotherapies [32,33].

Previous studies on host biomarkers for IPA investigated mainly genetic markers (single-nucleotide polymorphisms, SNPs [34,35]), cytokines, and already established infection and inflammation markers such as CRP [17,36]. In recent years, there has been an increasing amount of literature on cytokine levels in IPA patients, including elevated IL-6 and IL-8 levels in serum or BAL of aspergillosis patients [18,19]. Combinations of such host-derived data and multi-omics approaches have already been successfully applied in cancer research [37] and in several studies reporting promising results of using multi-biomarker signatures in differentiating viral from bacterial infection [38,39]. To the best of our knowledge, our pilot study is the first case-control study that investigated novel immune-relevant candidates indicating IPA in alloSCT patients using a multi-omics-level approach.

Our study revealed distinctive signatures associated with probable IPA cases compared with their controls consisting of *MMP1* induction and *LGALS2* downregulation together with elevated IL-8 and caspase-3 protein levels. While we evaluated the gene expression of selected candidates, including analyzing multiple additional in vitro expression data sets, protein levels of promising candidates were evaluated across two centers and thus additional patient samples (Figure 1). By exploring gene expression data sets of six relevant in vitro studies investigating host response following *A. fumigatus* exposure in human myeloid cells, we found supporting evidence for our transcriptome results focused on alloSCT patients for all investigated candidates with the exception of *CXCL8* (Appendix A). Our data indicate significant downregulation of *CXCL8* in cases compared with controls, however only with a low fold change difference and, thus, questionable clinical importance. MMP1 induction was demonstrated in all six investigated studies, while *LGALS2* repression was present in two studies using monocyte-derived dendritic cells. The latter can be explained by different in vitro models investigated in studies using different cell types, infection times, or different readouts (RNA-seq, microarrays).

Previous in vitro research demonstrated upregulation of *MMP1* following *A. fumigatus* exposure in different cell types [15,40], which is in line with our observations in whole blood of profiled probable IPA patients (Figure 3). Excessive amounts of MMP1 might lead to tissue destruction and help *A. fumigatus* to facilitate local invasion into blood vessels [15,41]. Galectin-2 (encoded by *LGALS2*) was reported to induce T-cell apoptosis and modulate inflammation through binding to the proinflammatory cytokine lymphotoxin-α [42,43]. Although its role during *Aspergillus* infections has not been evaluated, our study indicated promising *LGALS2* gene expression-based discrimination between probable IPA cases and their controls. More research is necessary to investigate whether *Aspergillus* spp. alter host *LGALS2* expression after infection and the exact role of galectin-2 in host response during IPA. We additionally demonstrated elevated caspase-3 and IL-8 serum levels in investigated IPA cases. IL-8 is considered to be one of the most promising biomarker candidates among the studied cytokines and its elevation in IPA patients has already been reported [18,19]. Following *Aspergillus* exposure, pro-caspases are induced via pattern recognition receptors, such as TLRs and dectin-1 [44]. Our results are in line with studies reporting caspase-3 induction as a response to *Aspergillus* infection and suggesting inhibition of caspase-3 as a pathogen strategy leading to inefficient host defence [22,45]. Furthermore, the induction of caspase-3 and IL-8 has recently been linked to fumagillin and gliotoxin in an in vitro model using a human epithelial cell line [46].

Although the EORTC/MSG criteria have been recently updated [2], the stratification of IPA patients remains challenging. Therefore, we further investigated MMP1, IL-8, and caspase-3 in possible IPA and CAPA patient sera to evaluate their potential as discriminative markers for different clinical manifestations of aspergillosis. We confirmed elevated IL-8 and caspase-3 levels in probable IPA patients in both hematological populations; nevertheless, this was not the case for possible IPA patients, reflecting the uncertainty of the diagnosis. While Cardiff possible IPA cases could be cases with false-negative mycology, Würzburg possible cases could actually reflect false-positive radiology. Our study thus highlighted the complexity of possible IPA cases by center-specific differences in the levels of the investigated molecular candidates, potentially due to different clinical characteristics of the included patients (underlying disease, IPA pathogenicity, treatment strategy), individual and subjective categorization of possible IPA cases in both centers, or the lack of specificity of the currently used diagnostic tools, emphasizing the need for further investigation in larger patient cohorts. To evaluate MMP1, IL-8, and caspase-3 in CAPA patients, we investigated additional COVID-19 patients developing aspergillosis as a secondary infection. COVID-19 patients presented different clinical characteristics compared with alloSCT patients (Appendix A). Interestingly, CAPA cases showed overall lower IL-8 and caspase-3 levels compared with COVID-19 controls (Figure 6), while in alloSCT patients IPA cases demonstrated elevated levels of the investigated candidates. These candidates, which are generally increased in COVID-19 patients [47,48], might indicate a risk for CAPA, given the fact that IL-8 and caspase-3 levels increase in hematological patients after *Aspergillus* infection (Figure 5). While reducing caspase levels in COVID-19 patients has recently been suggested as a treatment for severe COVID-19 [48], it has been demonstrated that, during fungal infection, caspase-3 inhibition limits cell apoptosis of alveolar macrophages, one of the crucial immune cells against *Aspergillus* infection [49]. As IL-8 and caspase-3 are usually abundant in COVID-19 patients, these findings encourage further evaluation of these candidates as potential risk factors for CAPA [47]. Moreover, our insights for the COVID-19 independent cohort highlighted the potential of IL-8 and caspase-3 in other clinical manifestations of aspergillosis. Since we followed a thorough pre-defined candidate selection process, using our original next to published data, further targets not investigated in this study may be promising. For instance, *PTX3* encoding pentraxin 3 has been reported to increase susceptibility to *A. fumigatus* infection in a *PTX3*^-/-^ mouse model [50] and was elevated in BAL samples of IPA patients suffering from pulmonary aspergillosis [51]. Although significantly different between IPA cases and controls using all our RNA sequencing samples, we did not observe a consistent separation across all time points, including the early phase after IPA onset and excluded this candidate from further analyses. Nevertheless, candidates such as *PTX3* arising from, e.g., our RNA sequencing data may be included in future studies.

We suggest an integrated patient-specific profile based on identified potential biomarker candidates. Combined with diagnostic assays and collected clinical information of profiled probable IPA cases, this molecular host component of probable IPA has the potential to substantially improve patient monitoring and therapy (Figure 7a). Although most of the combined features showed similar characteristics among all three patients, patient-specific profiles demonstrate challenges often met by clinicians due to the overall complexity of alloSCT patients and IPA (Figure 7a). While the treatment of patients defined with proven or probable IPA is justified by the weight of evidence underpinning the classification, in clinical practice clinicians are regularly faced with patients lacking typical radiology but with mycology suggestive of IPA or patients with radiology typical of IPA in the absence of supporting mycology (possible IPA). The lack of access to timely mycology drives empirical antifungals in high-risk patients with refractory fever. Given the toxicity and side-effects associated with antifungal therapy, the associated costs, and the emerging resistance to antifungal therapy, empiric administration of antifungal agents should be viewed critically. While the detection of the host-specific biomarkers of IPA will not necessarily pre-empt infection, it could enhance confidence in the diagnosis of IPA in patients with possible IPA or those with mycological evidence but not defined with IPA. Monitoring for these host-specific biomarkers could also play a role in determining patient prognosis. Currently, none of the fungal biomarkers have extensive evidence supporting this role and an integrative strategy incorporating fungal and selected new human biomarkers, in combination with clinical parameters, could prove optimal.

## 5. Conclusions

Despite the limited number of cases, our results, which are based on an integrative strategy for patient characterization using patient clinical information (e.g., antifungal treatment, etanercept administration, neutropenia, SNPs), diagnostic tests (*Aspergillus* PCR, Galactomannan assay, CT scans), and newly identified human biomarker candidates (*LGALS2*, *MMP1,* and caspase-3) (Figure 7b) are promising. Our results, therefore, encourage continued evaluation of this new integrated patient-specific approach using single human markers or a specific signature of markers in large multi-center studies for confirmation of our markers and profoundly reliable IPA monitoring. We envision that a patient-specific strategy combining fungal and human biomarkers might help clinicians to establish an individual antifungal treatment strategy or suggest close diagnostic monitoring of alloSCT patients with suspected IPA (Figure 7). The proposed strategic approach of integrating human biomarkers with the patient’s clinical information and already established fungal biomarkers might overcome current diagnostic deficiencies and therefore may contribute to improved diagnostic accuracy as well as therapeutic strategies of IPA.

## Figures and Tables

**Figure 1 jof-08-00171-f001:**
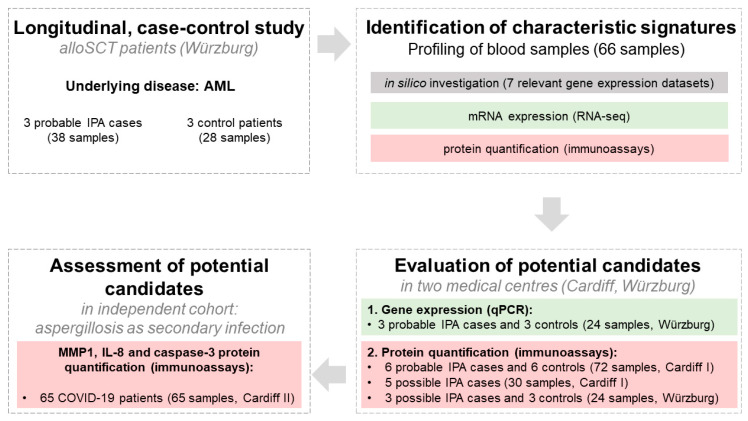
Workflow of a longitudinal case-control study investigating characteristic signatures indicating IPA among alloSCT patients. Characteristic signatures were identified by in silico investigation along with transcriptome and protein profiling of selected probable IPA cases and controls (Würzburg cohort). Potential molecular candidates were evaluated in additional alloSCT patients (Cardiff I, Würzburg) categorized as probable and possible IPA cases. The specificity of selected molecular candidates was investigated in independent cohorts consisting of CAPA patients and COVID-19 control patients (Cardiff II).

**Figure 2 jof-08-00171-f002:**
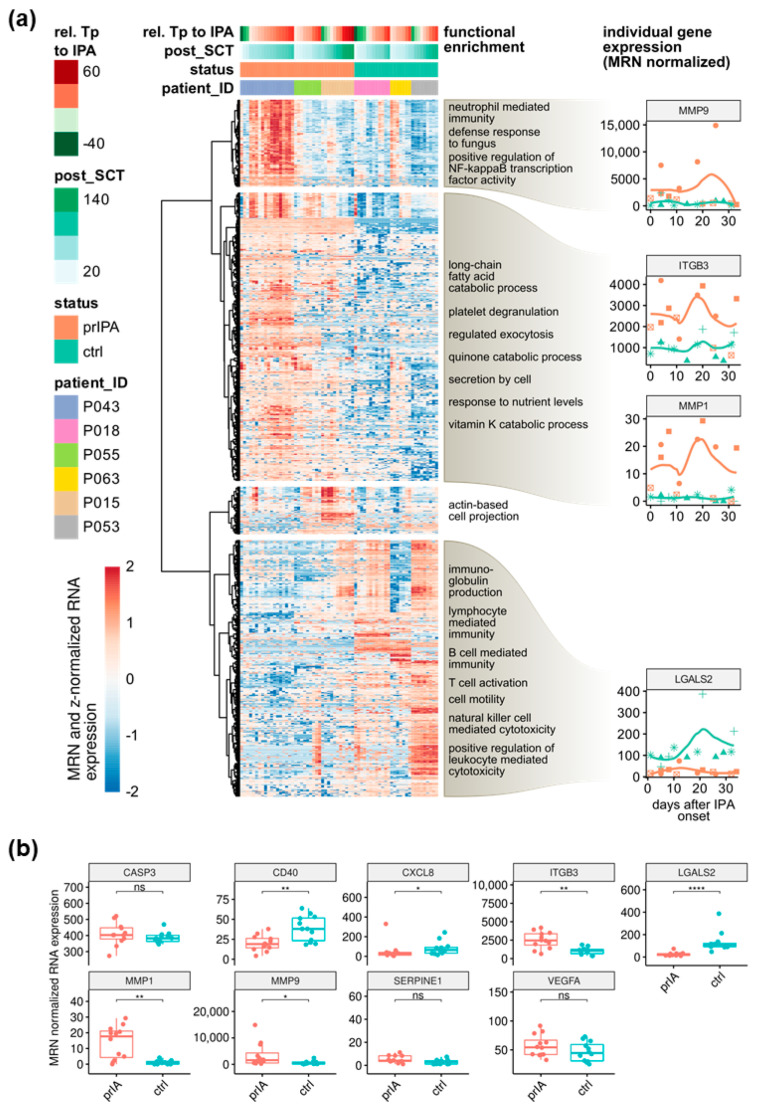
Transcriptome profiling of whole blood collected from three probable IPA patients and their matched controls. Blood samples were collected after alloSCT, twice-weekly whenever feasible and were investigated by RNA sequencing. (**a**) Hierarchical clustering and functional enrichment of genes differentially expressed across all collected blood samples in investigated probable IPA patients (red) and their matched controls (green) (*p* < 0.05, log2FC cutoff 0.8). Individual gene expression line plots refer to expression at days after IPA onset. (**b**) Median-ratio-normalized (MRN) RNA expression of selected candidates within the first 5 weeks after IPA onset (balanced dataset). Significance between IPA cases and control patients was tested by the Wilcoxon test (ns, not significant; *, *p* < 0.05; **, *p* < 0.01; ****, *p* < 0.0001).

**Figure 3 jof-08-00171-f003:**
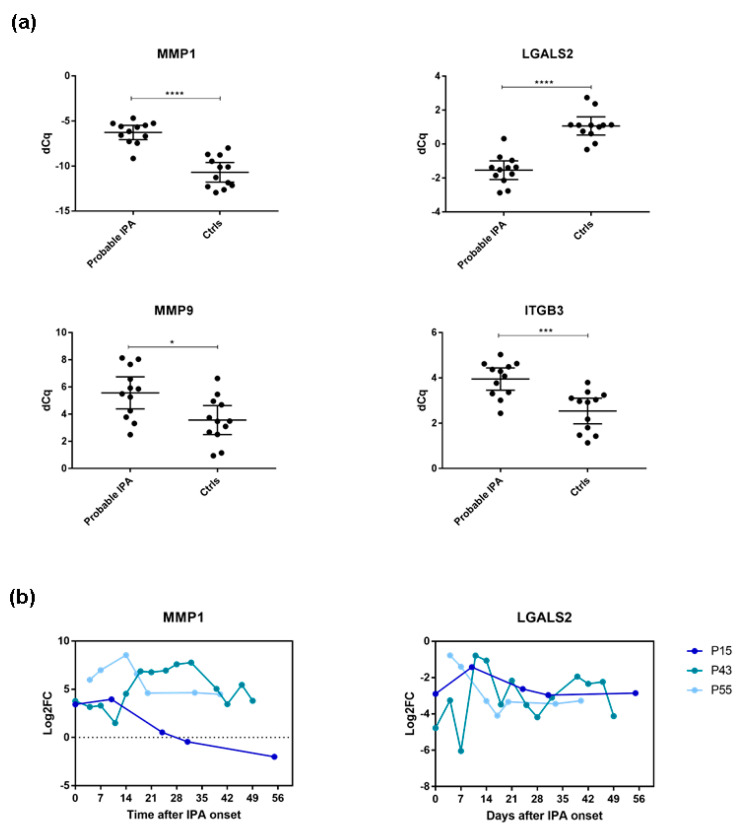
Gene expression of selected molecular candidates in probable IPA cases and their matched controls evaluated by real-time PCR. (**a**) Mean expression of *MMP1*, *LGALS2*, *MMP9*, and *ITGB3* in selected three probable IPA cases and their matched controls within 35 days after IPA onset (*n* = 24). Data are shown as mean ∆Cq values with 95% confidence intervals. (**b**) *MMP1* and *LGALS2* expression levels in probable IPA cases according to their matched controls within 56 days after IPA onset indicated as log_2_ of fold changes (log2FC). *n*, number of patient samples. Significant differences among investigated cases and controls were determined by the Mann–Whitney test (* *p* < 0.05, *** *p* < 0.001, **** *p* < 0.0001).

**Figure 4 jof-08-00171-f004:**
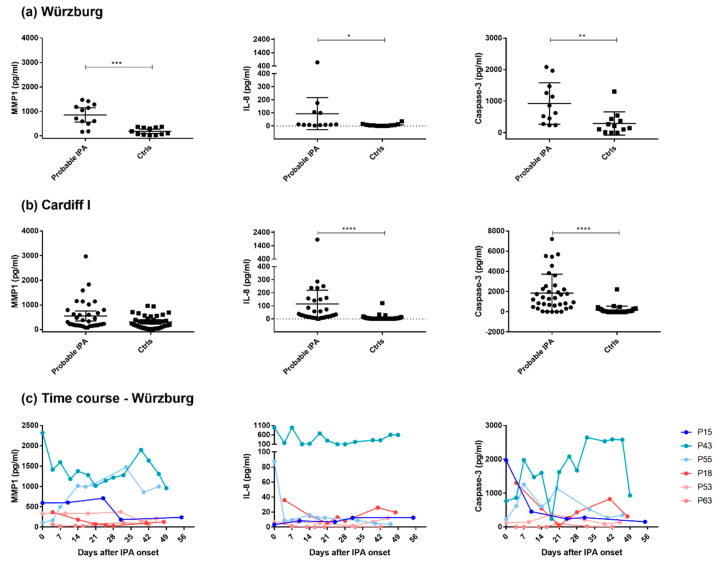
Serum protein levels of MMP1, IL-8, and caspase-3 investigated in probable IPA cases and their matched controls. Protein levels of potential candidates were investigated in (**a**) an original profiled patient cohort obtained from the University Hospital of Würzburg (*n* = 6), (**b**) a patient cohort (Cardiff I) obtained from Public Health Wales Microbiology Cardiff, United Kingdom (*n* = 12), and (**c**) original profiled patient cohort (Würzburg) samples collected up to 56 days after IPA onset (*n* = 6) (IPA cases in blue, control patients in red shades). Data in (**a**,**b**) are shown as mean protein levels with 95% confidence intervals. Significant differences among investigated cases and controls were determined by the Mann–Whitney test (* *p* < 0.05, ** *p* < 0.01, *** *p* < 0.001, **** *p* < 0.0001). *n*, number of patients.

**Figure 5 jof-08-00171-f005:**
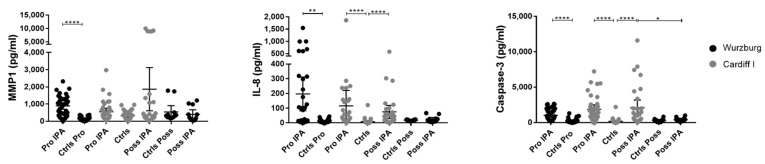
Investigation of MMP1, IL-8, and caspase-3 serum levels in additional alloSCT patients suffering from possible IPA (Poss IPA) and control patients (Ctrls), obtained from the University Hospital of Würzburg, Germany (black; N = 6; *n* = 24) and Public Health Wales Microbiology Cardiff, United Kingdom (grey; N = 11; *n* = 66). For direct comparison of possible IPA cases, probable IPA cases (Pro IPA) and matched controls (Ctrls) are added to the figure. Data are shown as mean values with 95% confidence intervals. Significant differences of investigated molecular candidates were investigated by the Mann–Whitney test (* *p* < 0.05, ** *p* < 0.01, **** *p* < 0.0001). N, number of patients; *n*, number of patient samples.

**Figure 6 jof-08-00171-f006:**
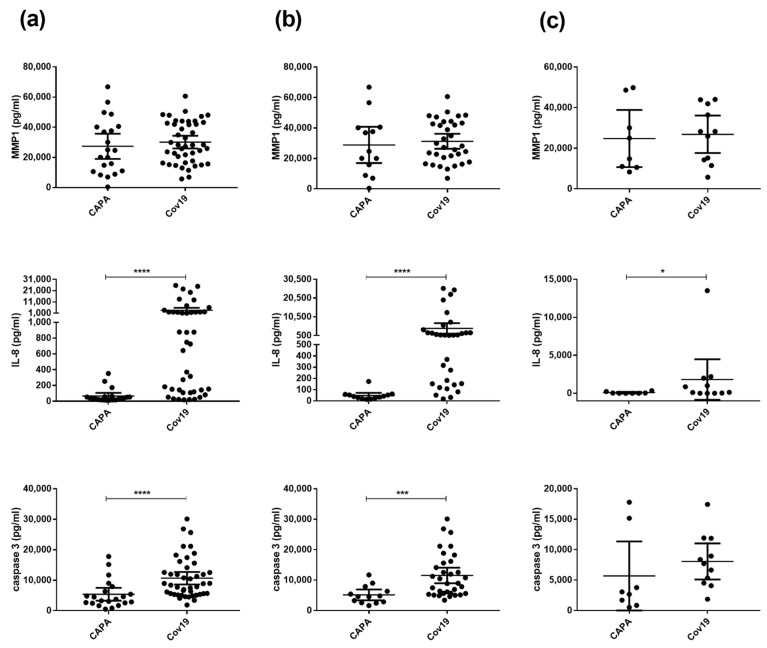
Protein quantification of MMP1, IL-8, and caspase-3 candidates in an independent patient cohort (Cardiff II) consisting of CAPA patients (CAPA) and control COVID-19 patients (Cov19) without Aspergillus infection (Cov19; N = 45). MMP1, IL-8, and caspase-3 serum protein levels are shown as mean values with 95% confidence intervals and were measured in (**a**) all investigated COVID-19 patients (N = 65), (**b**) in male COVID-19 patients (N = 46), and (**c**) in female COVID-19 patients (N = 19). The Mann–Whitney test was used to investigate significant differences among investigated CAPA and COVID-19 patients (* *p* < 0.05, *** *p* < 0.001, **** *p* < 0.0001). N, number of patients.

**Figure 7 jof-08-00171-f007:**
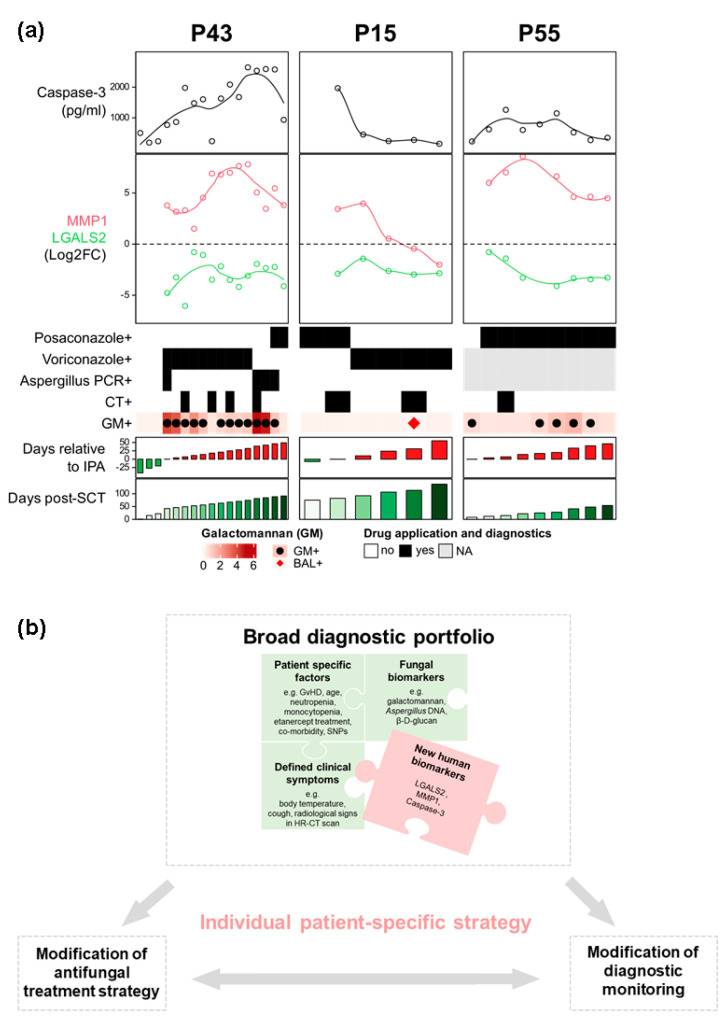
(**a**) Integrated biomarker-based profile of three investigated probable IPA cases and three controls. A proposed integrated profile consists of the suggested biomarker candidates (*LGALS2*, *MMP1*, and caspase-3), performed diagnostic tests (*Aspergillus* PCR, Galactomannan assay, and HR-CT), and collected clinical information (e.g., antifungal treatment, cell counts). Investigated time points with patients undergoing drug treatment and positive diagnostic tests are marked in black. In cases where HR-CT was not performed on the exact date of blood collection, results for positive CT are indicated closest to the investigated time point. GM+, Galactomannan positive for *A. fumigatus*; BAL, Bronchoalveolar lavage sample positive for *A. fumigatus*. Blood concentrations were center-scaled for comparability. (**b**) Proposed individual patient-specific strategy based on the combination of human and fungal biomarkers with patient-specific features.

**Table 1 jof-08-00171-t001:** Clinical information of profiled IPA cases and their matched controls (original cohort, Würzburg).

Patient	Status	Age	Sex	Antifungal Treatment	Prophylaxis	GvHD (Grade)	No. of Samples
P15	case	63	M	VRC	PSC, FLC	pulmonary (III)	11
P53	control	59	M	NT	FLC	no GvHD	9
P43	case	60	F	VRC, PSC	FLC	skin (III), intestinal (IV)	18
P18	control	51	F	NT	PSC	no GvHD	12
P55	case	62	M	Ambisome	PSC	skin (NA)	9
P63	control	55	M	NT	FLC	skin (III) and intestinal (I)	7

M, male; F, female; VRC, voriconazole; PCS, posaconazole; FLC, fluconazole; NA, not known; NT, no antifungal treatment.

## Data Availability

The data discussed in this publication have been deposited in NCBI’s Gene Expression Omnibus (GEO) and are accessible through GEO Series accession number GSE174825 (https://www.ncbi.nlm.nih.gov/geo/query/acc.cgi?acc=GSE174825).

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
