# Peer review of "Molecular Profiling Reveals Characteristic and Decisive Signatures in Patients after Allogeneic Stem Cell Transplantation Suffering from Invasive Pulmonary Aspergillosis"

_jof, 2022, doi:10.3390/jof8020171_

Round 1
Reviewer 1 Report
I commend the authors for their extensive work. Many work has been put to bring all these data together. The only question I raise is why PTX3 has not been tested.
Reviewer 2 Report
Nowaday diagnosis of pulmonary diseases which releated to molds, are current topic. This article can help in diagnosis of invasive pulmonary aspergillosis in patients.
Author Response
Reviewer 2: Nowaday diagnosis of pulmonary diseases which releated to molds, are current topic. This article can help in diagnosis of invasive pulmonary aspergillosis in patients.
Response: We thank the reviewer for the positive judgement of our manuscript.
Reviewer 3 Report
The paper entitled "Molecular profiling reveals characteristic and decisive signatures in patients after allogeneic stem cell transplantation suffering from invasive pulmonary aspergillosis" by Tamara Zoran et al describes the challenging task of host markers as a tool for pathogen diagnostics.
Obviously there is still a long way to go for an ultimate pathogen-specific diagnostics which is based on host markers, if possible at all. The authors are not claiming to provide such an argument. The context of the data in the current paper is is as a support for integrated clinical decision making.
The paper is well written, data are clearly presented and I would definitely recommend to publish it in your journal.
Please note, in line 540 the comma is misplaced.
Good luck
Author Response
Reviewer 3: The paper entitled "Molecular profiling reveals characteristic and decisive signatures in patients after allogeneic stem cell transplantation suffering from invasive pulmonary aspergillosis" by Tamara Zoran et al describes the challenging task of host markers as a tool for pathogen diagnostics. Obviously there is still a long way to go for an ultimate pathogen-specific diagnostics which is based on host markers, if possible at all. The authors are not claiming to provide such an argument. The context of the data in the current paper is is as a support for integrated clinical decision making.
The paper is well written, data are clearly presented and I would definitely recommend to publish it in your journal.
Response: We appreciate that the reviewer recognized the value of our study and the challenges present when working with patient data. We are very grateful for the positive judgment of our work.
Reviewer 3: Please note, in line 540 the comma is misplaced.
Response: As there was no comma in line 540, we assumed a typo and corrected line 450. Thank you for the remark.